# Reconstructing the population history of the sandy beach amphipod *Haustorioides japonicus* using the calibration of demographic transition (CDT) approach

Kay Sakuma *°, Risa Ishida°, Taketoshi Kodama°, Yoshitake Takada°

Japan Sea National Fisheries Research Institute, Fisheries Research and Education Agency, Niigata, Japan

° These authors contributed equally to this work.

* keisakuma@affrc.go.jp

**Data Availability Statement:** All nucleotide sequences are available from GenBank (accession numbers LC474498–LC474506, also see S1 Table).

## Abstract

Calibration of the molecular rate is one of the major challenges in marine population genetics. Although the use of an appropriate evolutionary rate is crucial in exploring population histories, calibration of the rate is always difficult because fossil records and geological events are rarely applicable for rate calibration. The acceleration of the evolutionary rate for recent coalescent events (or more simply, the time dependency of the molecular clock) is also a problem that can lead to overestimation of population parameters. Calibration of demographic transition (CDT) is a rate calibration technique that assumes a post-glacial demographic expansion, representing one of the most promising approaches for dealing with these potential problems in the rate calibration. Here, we demonstrate the importance of using an appropriate evolutionary rate, and the power of CDT, by using populations of the sandy beach amphipod *Haustorioides japonicus* along the Japanese coast of the northwestern Pacific Ocean. Analysis of mitochondrial sequences found that the most peripheral population in the Pacific coast of northeastern Honshu Island (Tohoku region) is genetically distinct from the other northwestern Pacific populations. By using the two-epoch demographic model and rate of temperature change, the evolutionary rate was modeled as a lognormal distribution with a median rate of 2.2%/My. The split-time of the Tohoku population was subsequently estimated to be during the previous interglacial period by using the rate distribution, which enables us to infer potential causes of the divergence between local populations along the continuous Pacific coast of Japan.

## Introduction

Calibration of the molecular clock is one of the major challenges in marine population genetics. It is undoubtedly crucial to use an appropriate evolutionary rate to obtain demographic timelines as well as population parameters such as divergence time and migration rates [1, 2]. Although fossil records are most commonly used in phylogenetic rate calibration, they may

**Funding:** This study was supported by "A project for development of assessment methods for the coastal environment" funded by the Ministry of Agriculture, Forestry and Fisheries, and Grants-in-Aid for Scientific Research from the Ministry of Education, Culture, Sports, Science and Technology, Japan (25340114 and 15H02265). There was no additional external funding received for this study.

**Competing interests:** The authors have declared that no competing interests exist.

not be applicable in population-level studies because little or no morphological difference among fossils is expected for younger episodes [3]. Geological dating is also unrealistic for some marine species because complete separation of populations is sometimes implausible because of gene flow among populations via passive transport (e.g., larval and egg dispersal) or active migration over long distances [4]. Phylogenetic evolutionary rates of related species are sometimes applied, but this is controversial because of the variation in molecular rates among organisms [5, 6].

Acceleration of the evolutionary rate is the other source of errors in molecular dating, and is particularly emphasized in population studies. Ho et al. [7] first summarized ideas about rate change by modeling the rate as an exponential function of time since calibration events and empirical studies subsequently verified this behavior in molecular clocks [8–10] (reviewed in [3]). This "time-dependency of the molecular clock" is partly explained by selection and genetic drift after deep phylogenetic events, which leads to reduction of the molecular rate [3, 7]. Grant [2] recommended caution in using a phylogenetic rate for population studies, especially for demographic reconstruction, because it can causes overestimation of the coalescence time, leading to misinterpretation of the results.

Calibration of demographic transition (CDT) [11] is one of the most promising techniques currently available for overcoming difficulties in rate calibration. CDT is a generalization of expansion dating [9] that assumes demographic expansion for some shallow-water invertebrate species in relation to an increase in habitat availability after the last glacial period. Both CDT and expansion dating use a two-epoch coalescent model (TEM), which assumes an ancient epoch of constant population size followed by a modern epoch of rapid population increase [12]. In expansion dating, the initial dates of the habitat increase (19.6 and 14.6 kya, based on two different assumptions) was used to obtain deterministic estimates of the evolutionary rate. CDT also assumes a post-glacial expansion but uses temperature as a demographic proxy in a calibration of species genealogy and obtains a stochastic function for the evolutionary rate. CDT is applicable for a wide range of species [11]. Regardless of its potential usability very few studies have used CDT, probably because its power in marine population genetics is currently not widely known.

We focused on the sandy beach amphipod *Haustorioides japonicus* Kamihira, 1977 (Amphipoda: Dogielinotidae) in the present study as a model for exploring population history because its biological traits potentially allow for reconstruction of the population history from molecular data. This species is characterized by an extremely low rate of effective migration and thus limited gene flow between local populations [13] because of the lack of a planktonic dispersal stage [14] and its specific habitat requirement for sandy beaches [15]. The species is distributed along the Sea of Japan and Pacific coasts of Hokkaido Island, but has also been recorded in the Tohoku region along the Pacific coast of northeastern Honshu Island, Japan (Fig 1) [16]. Considering the extremely low migration activity of the species, the populations in the Tohoku region are assumed to be genetically distinct from other known populations, with a local population history that is linked to the paleoceanography of the area.

Here we incorporate CDT with a Bayesian Skyline Plot (BSP) [17] for the reconstruction of genealogy of the sandy beach amphipod *Haustorioides japonicus*, and demonstrate the importance of an appropriate molecular rate calibration in marine population studies. We exhaustively sampled the sandy beach amphipod in the Tohoku region to find the most peripheral populations of the species and assigned them to currently recognized sandy beach amphipod populations (Fig 1) [13]. We then inferred population histories by reconstructing demographic timelines, estimating the split-time of the local populations, and considering the biological traits of the species and the paleoceanography of the area.

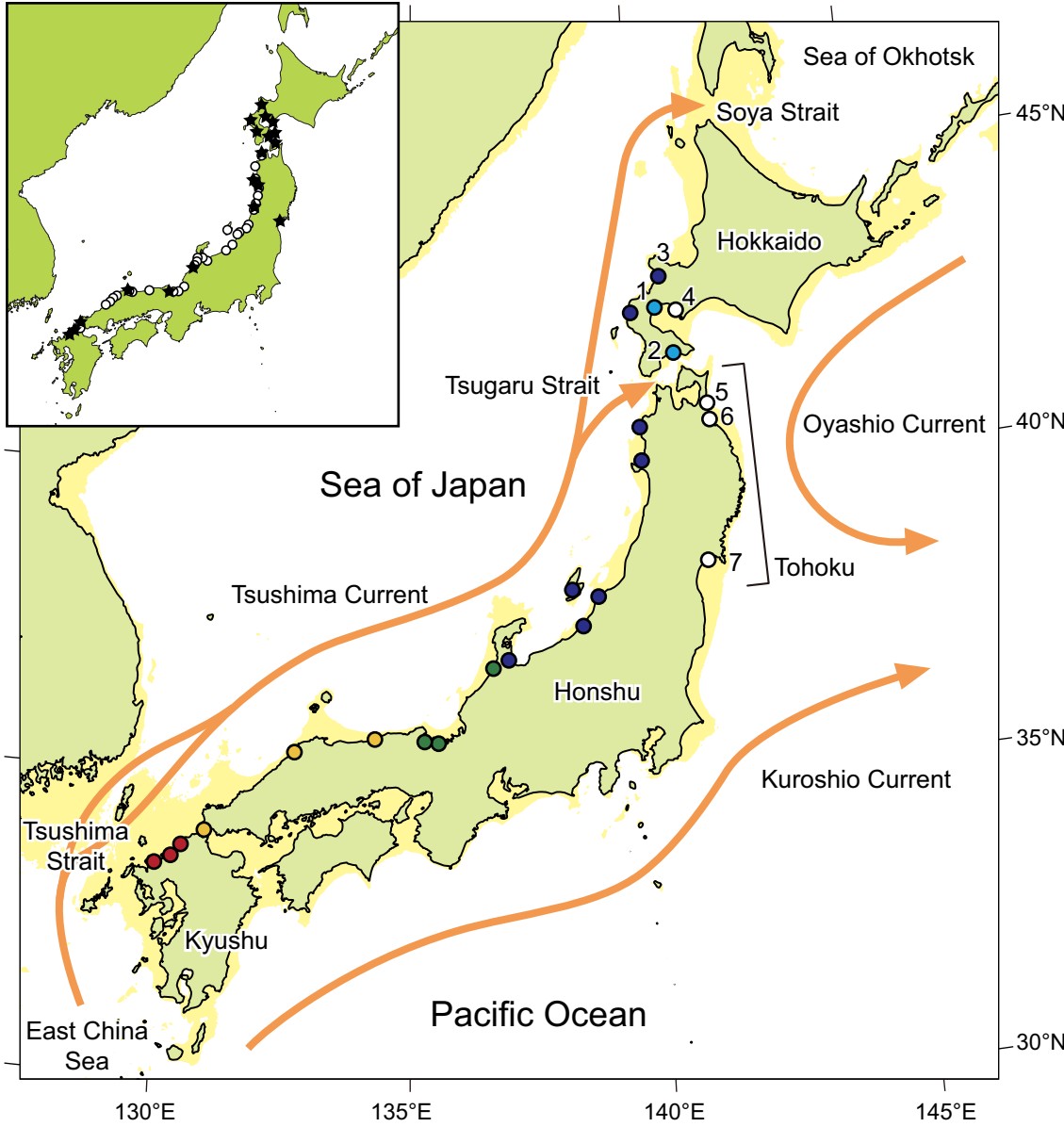

**Fig 1. Map showing sampling sites for the sandy beach amphipod *Haustorioides japonicus*.** Empty circles and numerals show locations and ID numbers for sampling sites in the present study. Filled and colored circles indicate the population assignment of the sites in a previous study (Takada et al. 2018 [13]; see also Fig 2). Shaded areas indicate the coastlines during the last glacial maximum (−120 m). Map insert shows previously reported sites with *H. japonicus* (stars, Kamihira 2000 [16]; circles, Takada et al. 2015 [15]). The map was created with QGIS v2.18.0 (http://www.qgis.org) using layers freely available at Natural Earth (https://www.naturalearthdata.com/downloads/10m-physical-vectors/).

## Materials and methods

### Field surveys and sample collection

A total of 32 specimens were collected from three sandy beaches in the Tohoku region and one site along the coast of Hokkaido (Fig 1, Table 1). Specimens were collected by using a 1-mm-mesh sieve, and parts or whole bodies of individuals were preserved in 6 M TNES (Tris HCl, EDTA, NaCl, SDS) urea buffer [18] (hereafter, "urea buffer"). The urea buffer contains a high

**Table 1. Sampling location, sample size (*n*), coordinates, collection date, and reference for sandy beach amphipod *Haustorioides japonicus* specimens used in this study.**

| ID | Location | *n* | Latitude | Longitude | Date | Reference |
|----|----------|-----|----------|-----------|------|-----------|
| 1 | Oshamanbe | 24 | 42.526 | 140.393 | May 2015 | Takada et al. [13] |
| 2 | Hakodate | 20 | 41.770 | 140.743 | May 2015 | Takada et al. [13] |
| 3 | Tomari | 3 | 43.031 | 140.524 | May 2015 | Takada et al. [13] |
| 4 | Usu | 11 | 42.512 | 140.780 | May 2015 | This study |
| 5 | Ogawara | 3 | 40.923 | 141.393 | July 2018 | This study |
| 6 | Misawa | 3 | 40.674 | 141.439 | July 2018 | This study |
| 7 | Shichigahama | 15 | 38.289 | 141.066 | July 2018 | This study |

concentration of urea, which allows cell lysis and DNA extraction and preservation at ambient temperature. Each specimen was identified to species in the laboratory according to Kamihira [14] using a stereomicroscopy.

## Laboratory procedures and data processing

Total DNA was extracted from the urea buffer-tissue lysate using a DNeasy Blood & Tissue Kit (Qiagen Inc., Valencia, CA, USA) according to the manufacturer's instructions. Partial sequences of the mitochondrial cytochrome *c* oxidase subunit I (COI) gene were amplified by PCR using the primers LCO1490 and HCO2198 [19] (for details, see Takada et al. [13]). The products were directly purified by ExoSAP-IT (Affymetrix, Santa Clara, CA, USA) and sequenced in both directions at Food Assessment & Management Center (FASMAC) Inc. (Kanagawa, Japan).

The forward and reverse sequences were assembled by using Mesquite v3.01 [20] and visually inspected (accession numbers: LC474498–LC474506, also see S1 Table). The 105 sequences used by Takada et al. [13] (accession numbers LC224174–LC224278) were added to the dataset and multiple sequence alignment was subsequently performed using MAFFT v7.294 [21]. The best nucleotide substitution model for the dataset was determined using MEGA7 [22]. The sequence dataset was then collapsed into haplotype using custom Perl script.

To assign sequences from the Tohoku region to currently known sandy beach amphipod populations, we constructed a minimum spanning tree using the software packages Arlequin v3.5 [23] and SplitTree v4 [24]. Genetic differentiation between the populations was confirmed by the pairwise Φst test. We also obtained standard genetic diversity indices (*H*, number of haplotypes; *h*, haplotype diversity; *π*, nucleotide diversity) and neutrality indices (Tajima's *D* [25]; Fu's $F_s$ [26]; $R_2$ [27]). These statistical tests were performed with Arlequin except for $R_2$ test done on DnaSP v6 [28]; significance of the pairwise Φst and neutrality indices was assessed with 1000 permutations.

## Rate calibration using the CDT approach

Rate calibration was performed following standard CDT procedures [11], but with some minor modifications (hereafter 'rate' means substitution rate, not 'divergence' rate). To minimize coalescent errors and improve convergence, we built TEMs using all individuals from Tohoku as well as its parental population among currently known sandy beach amphipod populations. We created custom .xml files for BEAST v1.7 [29] for the combination of two clock models (strict clock, log-normal and exponential relaxed) and two TEMs (exponential and logistic; also see S1 Fig), referring to the example file of Hoareau [11] (Dryad Digital

Repository; http://dx.doi.org/10.5061/dryad.3q24t). A total of 10,000 out of $5.0 \times 10^8$ Markov chain Monte Carlo (MCMC) steps were recorded by using BEAST. The results from the six models were then compared by using the Bayes factor test implemented in Tracer v1.6 [30]. After discarding the first 1000 records (i.e., 10%) as a burn-in, 9000 values of the mutation-scaled transition time from the best model were obtained. We subsequently downloaded 10,000 calendar years derived from the discretized change rate of temperature over the last 25 ky (Dryad Digital Repository; see Jouzel et al. [31] for original data) to obtain $9 \times 10^7$ mutation-rate values. The density distribution of the mutation rate was then modeled using log-normal and Gaussian functions, and the best model was selected based on the Akaike information criterion (AIC). These statistical calculations were performed using MASS v7.3 package [32] in R v3.4.3 [33].

### Reconstruction of population history

We inferred the demographic history of the Tohoku and its parental population using a BSP [17], and the genealogy derived from the Bayesian skyline model. We first built a BSP with a total of 10,000 records out of $2.0 \times 10^9$ MCMC steps and discarded the first 1000 records (i.e., 10%) as a burn-in. The number of groups was set to 10 and a piecewise-linear BSP model was adopted. The molecular rate was modeled following the best clock model and statistical distribution from CDT. We also built a BSP under a strict clock model with a conventional evolutionary rate for crustacean mitochondrial genes of 0.7%/My [34] as a proxy of 'general' rate to compare results from different molecular clock assumptions. The site model was set to the best model inferred by MEGA. A generation time of one year [14] was assumed to convert generation time-scaled female effective population size into effective population size ($N_e$). The split-time of the Tohoku population from the parental population was subsequently estimated based on the genealogy of the mitochondrial COI and reconstructed along with the demographic changes in the BSP. TreeAnnotator v1.7 [28] was used for time-tree reconstruction. We used 9000 trees after discarding the first 1000 trees out of 10,000 and obtained a maximum clade credibility tree with a median node height. The posterior distribution of the split-time between the Tohoku and northwestern Pacific (NWP) populations was then summarized using TreeStat v1.7 [28].

### Results

All of the 32 newly-obtained sequences were assigned to a group that is common along the coast of the northwestern Pacific Ocean (Fig 2), and the following analyses were therefore performed on 79 sequences for this genetic group, including the 47 previously reported (Table 1). The final dataset comprises 608 bp of COI sequences from 79 individuals, which were collapsed into 15 haplotypes defined by 17 polymorphic sites. The best-fit model was estimated to be HKY + Γ [35, 36]. Frequencies and accession numbers for the 15 haplotypes are given in S1 Table.

Among 79 individuals of the NWP group, 15 from the southern Tohoku region (site 7, Fig 1) are genetically distinct from the other members of the NWP group ($\Phi_{st} = 0.75$, $P < 0.05$; Fig 2). We therefore treat the group of 15 individuals from site 7 as a distinct population (hereafter "Tohoku population") and we explore its history in the following section. Table 2 includes summary statistics for the two group of populations (NWP and Tohoku). Neutrality indices for these populations were negative and significant except for the $R_2$ value of Tohoku population.

A logistic TEM with exponential relaxed clock (model 3, Table 3) was chosen as the best model in the Bayes factor test, although there was no substantial difference between median

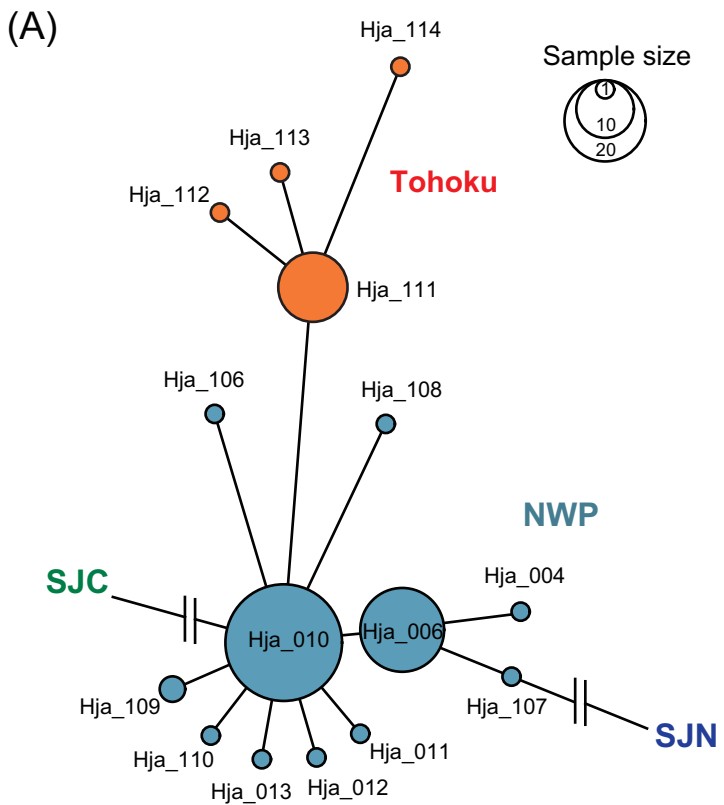

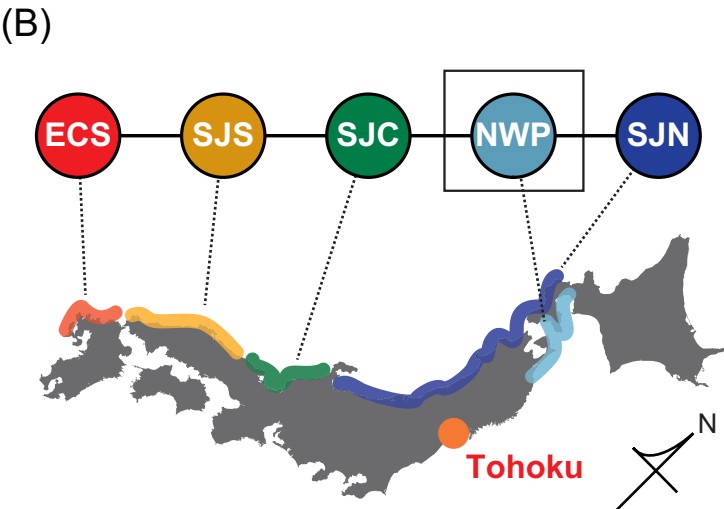

**Fig 2. Minimum spanning tree of mitochondrial COI haplotypes from the northwestern Pacific (NWP) population of the sandy beach amphipod *Haustorioides japonicus* (A), and interrelationships among and geographic distributions of *H. japonicus* populations (B; this study and Takada et al. [13]).** Branch length and circle size are proportional to the T92 + Γ distance and haplotype frequency, respectively. The haplotype ID is beside each node (see also S1 Table). ECS, East China Sea; SJS, southern Sea of Japan; SJC, central Sea of Japan; NWP, northwestern Pacific; SJN, Northern Sea of Japan.

**Table 2. Genetic diversity and neutrality indices for two group of populations of the sandy beach amphipod *Haustorioides japonicus*.**

| Population | Standard diversity indices | | | | Neutrality indices | | |
|---|---|---|---|---|---|---|---|
| | *N* | *H* | *h* | *π* | *D* | *F*$_s$ | *R*$_2$ |
| NWP | 64 | 11 | 0.62 ± 0.05 | 0.0014 ± 0.0011 | -1.92* | -7.48* | 0.042* |
| Tohoku | 15 | 4 | 0.37 ± 0.15 | 0.0009 ± 0.0009 | -1.82* | -1.72* | 0.143 |

*n*, number of individuals; *H*, number of haplotypes; *h*, haplotype diversity; *π*, nucleotide diversity; *D*, Tajima's *D*; *F*$_s$, Fu's *F*$_s$. Asterisks (*) indicate significance of tests at *P* < 0.05.

estimates of transition time in the four models. For the exponential TEM with strict clock, we could not obtain the result because the runs did not converge. Hereafter we used the best TEM only. The density distribution of transition time was unimodal and the median estimate was $2.58 \times 10^{-4}$ mutations/site in the TEM (Fig 3). We selected the log-normal model, which had a lower AIC than the Gaussian model for the density distribution of evolutionary rate (S1 Fig); LogMean and LogSD for the log-normal approximation were –3.89 and 0.53, respectively. The median CDT rate was 2.2%/My.

The BSP was built with exponential relaxed clock using all sequences from NWP and Tohoku populations and the CDT rate, revealed rapid growth after the last glacial period, coinciding with warming temperatures, whereas the time of demographic expansion was estimated to be during the last glacial maximum (20–40 kya) with a phylogenetic rate of 0.7% (Fig 4). The divergence between NWP and Tohoku populations falls back to the previous interglacial period (median = 97 kya, but with a 95% credibility interval of 11–360 kya; Fig 5) using the CDT rate, whereas the estimate is 340 kya (95% credibility interval: 61–450 kya) based on the phylogenetic rate.

## Discussion

It is generally difficult to explore the histories of marine populations because vicariant events and fossil records are rarely applicable to the rate calibration. Even if these calibration techniques happen to be available for a target species, they can cause misinterpretation of the results because of the time dependency of the molecular clocks [2]. The use of demographic transition as a global rate calibration point enables us to overcome these difficulties in marine population genetics and attribute recent coalescent events such as population splits to certain geological episodes. In the following section, we elucidate the history of the sandy beach amphipod population and compare our results from the CDT approach with those based on a phylogenetic rate. We then discuss the limitations of the current approach.

**Table 3. Model comparison results.** Listed are the model ID, type of two-epoch model (TEM) and clock model, transition time (TT, $10^{-3}$ mutations/site), median marginal likelihood (MML), and 2×log Bayes factors (2×lnBF) for the models. Parameters for the best-fit model are shown in bold type. The result for the exponential TEM with strict clock is not shown because the run did not converged.

| Model | TEM | Clock model | TT | MML | 2×lnBF | | | | |
|---|---|---|---|---|---|---|---|---|---|
| | | | | | #1 | #2 | #4 | #5 | #6 |
| 1 | Exponential | Exponential | 2.60 | -1012.3 | | 7.2 | -6.8 | -4.8 | 8.0 |
| 2 | Exponential | Lognormal | 2.51 | -1015.9 | -7.2 | | -12.0 | -10.0 | -8.0 |
| **3** | **Logistic** | **Exponential** | **2.56** | **-1008.9** | **6.8** | **14** | | **2.0** | **14.8** |
| 4 | Logistic | Lognormal | 2.53 | -1009.9 | 4.8 | 12.0 | -2.0 | | 12.8 |
| 5 | Logistic | Strict | 2.59 | -1016.3 | -8.0 | -0.8 | -14.8 | -12.8 | |

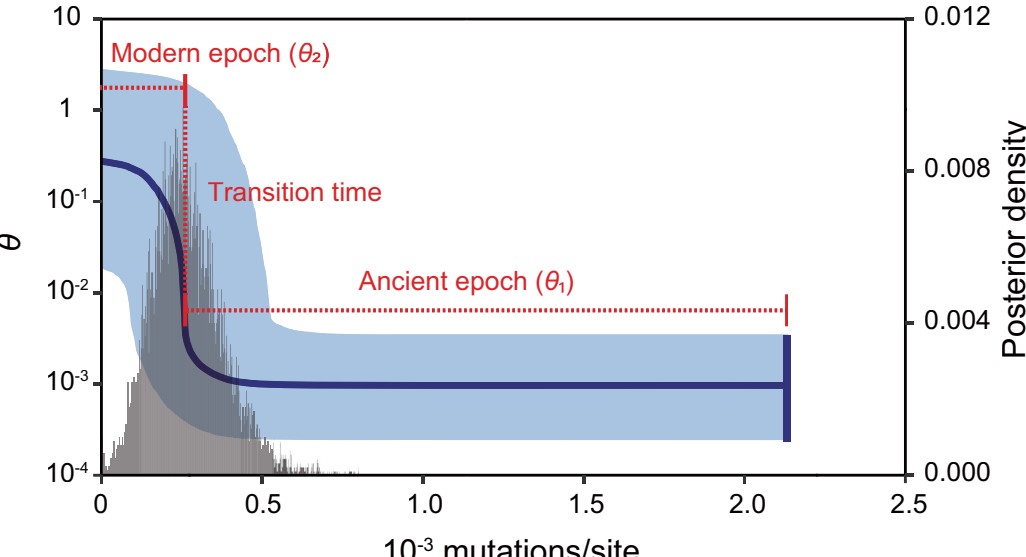

**Fig 3. Demographic timeline derived from the best two-epoch demographic model (model 3, Table 3).** An ancient epoch with a constant population size ($\theta_1$) is followed by the modern epoch of logistic population growth ($\theta_2$). The median estimate of the transition time is shown as a vertical red line. Horizontal and vertical blue lines and the blue shaded area show the median estimate for mutation-scaled female effective population size ($\theta$), the time of the most recent common ancestor, and the 95% credible interval, respectively. The histogram shows the posterior density distribution for the two-epoch transition time.

### Fluctuations of temperature as a proxy for demography

We observed a strong concordance between fluctuations in temperature and historical demography of the sandy beach amphipod population (Fig 4). Post-glacial expansion in the NWP is

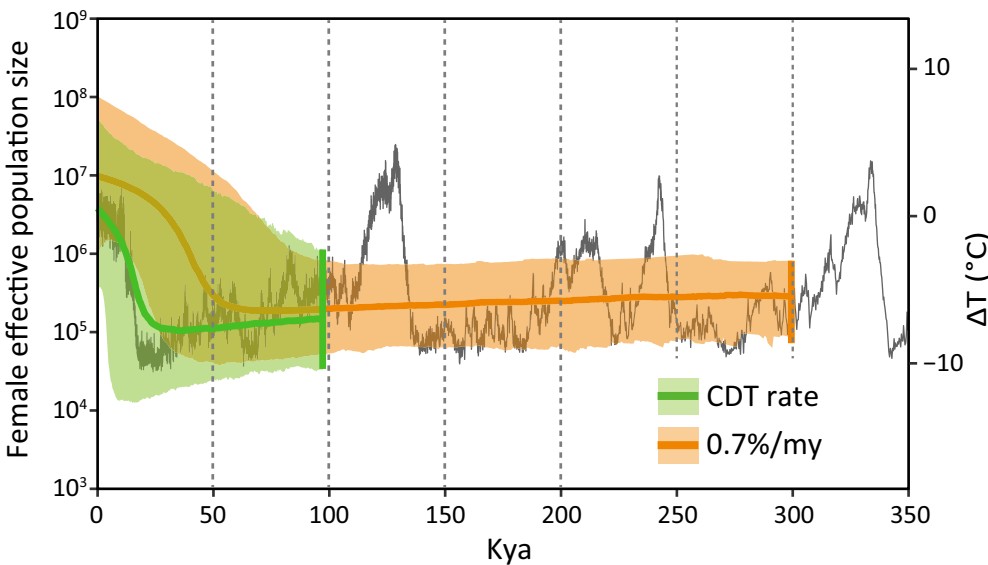

**Fig 4. Bayesian skyline plots based on the calibration of demographic transition (CDT) rate and a conventional mitochondrial evolutionary rate of 0.7%.** Median estimates of female effective population size are shown as lines and vertical lines show median estimates of the time of the most recent common ancestor. Global trends of historical temperature are superimposed on the plots (Jouzel et al. [31]).

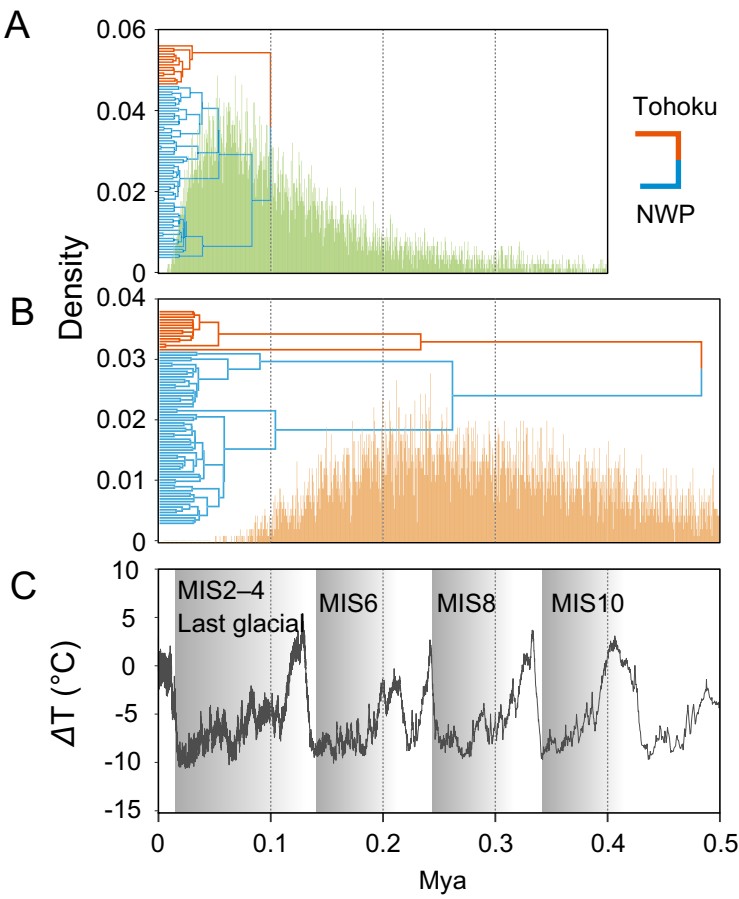

**Fig 5. Posterior density distribution histograms of the split-time between Tohoku and other NWP individuals estimated by using CDT rate (A) and a conventional mitochondrial evolutionary rate of 0.7% (B), and global trends of historical temperature from Jouzel et al. [31] (C).** Maximum clade credibility trees are also shown for each rate.

much more plausible than population growth throughout the last glacial maximum, considering the life-history traits of the species and the paleoceanography of the northwestern Pacific Ocean. The sandy beach amphipod population grows rapidly in late spring to summer from a small number of overwintering individuals [37]. The extended winter season during the last glacial period probably shortened the reproductive season for the species, which could have caused a serious decrease in effective population size. Sea-ice cover in the area expanded both spatially and temporally during the last glacial period in this region [38], which could have also altered the environmental conditions on sandy beaches. After the last glacial period, the population seems to have rapidly recovered from the severe reduction in size in parallel with warming temperatures (Fig 4). Our results from the CDT rate seem quite reasonable considering that the North Atlantic intertidal gastropod *Littorina saxatilis* Olivi, 1792 whose habitat conditions are close to those of the sandy beach amphipod, also showed post-glacial expansion [11].

### Divergence between the local populations

In the NWP, sea-level changes throughout glacial cycles have greatly influenced marine populations via changes in both inter-population connectivity and oceanographic conditions [39–43]. Because the Tsugaru Strait connecting the NWP and the Sea of Japan was narrow and

shallow during the last glacial period, the Tsushima Current from the Sea of Japan (Fig 1) ceased [44], possibly causing lineage splits for marine species. Considering that the posterior distribution of the split-time shows a peak during the last glacial period (Fig 5), it can be assumed that gene-flow among the sandy beach amphipod populations was lost during the last glacial period as a result of the change in oceanic currents, resulting in genetic differentiation of the Tohoku population (Fig 1).

Alternatively, we also hypothesized that occasional migration events contributed to the southward range expansion of the species. We note that tsunamis are one of the possible transport mechanisms in the Tohoku region because they have repeatedly swept sandy beaches in this area [45]. Most recently, a megathrust M9.0 earthquake generated a huge tsunami on 11 March 2011, which caused serious damage in shallow-water environments and greatly altered benthic community structure in this region [46–49]. We discovered, however, that the local sandy beach amphipod population in the Tohoku region persisted after the tsunami, which strongly suggests the survival or resettlement of the local population. It is thus plausible that there were occasional migrations between local sandy beaches driven by episodes such the Great Tsunami. Unfortunately, it is difficult to attribute a population split to a specific geological event, owing to the large credible interval (Fig 5). Increasing the number of individuals, loci, and sites sampled would improve the precision of the split-time estimates and is thus a promising avenue for further exploration of the population history of this species.

## Limitations of the demographic rate calibration for the sandy beach amphipod

Rapid growth of a population after the last glacial period is a fundamental assumption of CDT [11]. Although we have discussed the plausibility of the post-glacial expansion hypothesis for this species in the previous section, there still remains some uncertainty. The CDT rate from the present study exceeds the phylogenetic rate of 1%, but is much slower than that from Hoareau [11], possibly signaling an underestimation of the rate. We also note that there is a range among observed rate for the crustacean species [9, 50, 51]. The difficulties are mainly caused by the fact that ultimately it is impossible to know the real trigger for the population expansion. A false assumption of post-glacial expansion is a potential limitation of the demography-based rate calibration technique [9, 11], and this possibility must always be considered.

Additional sources of uncertainty in BSP and estimation of the divergence time are large coalescent errors caused by a small number of sequences, loci, and polymorphic sites [2]. We used only a single locus of COI with 608 bp from 79 individuals, which could lead to misinterpretation of the results. Also, using the probabilistic CDT approach increases the range of the credible interval for the BSP and split-time, which makes it difficult to interpret the results, as discussed in the previous section. The two-epoch demographic model is currently applicable only for a single gene in BEAST, and this becomes a more practical problem when one wishes to improve the credibility of the analysis by using a larger dataset. The implementation of multi-locus data in the program will help to deal with this issue.

## Conclusion

We have demonstrated the applicability of rate calibration with global demographic transition by using the CDT approach. The median estimate of the CDT rate was approximately 2%/My for COI sequences from the sandy beach amphipod, which exceeds the conventional phylogenetic rate of 0.7%/My. The BSP for the entire population was more realistic using the CDT rate than the phylogenetic rate, which detected population growth throughout the last glacial maximum, in terms of the post-glacial expansion (Fig 4). We further inferred the drivers of the

population split by estimating a divergence time from the CDT rate. Expanding the dataset will enable further exploration of the population history of marine organisms such as the sandy beach amphipod.

## Supporting information

**S1 Fig. Histogram showing the frequency distribution of evolutionary rate derived from CDT.** Log-normal and Gaussian approximations of the rate are also shown.
(PDF)

**S1 Table. List of haplotypes.** Haplotype ID, total number of individuals ($n$), number of individuals per site (Per site), and accession number are shown.
(PDF)

## Author Contributions

**Conceptualization:** Kay Sakuma, Yoshitake Takada.

**Data curation:** Kay Sakuma, Risa Ishida.

**Formal analysis:** Risa Ishida.

**Funding acquisition:** Taketoshi Kodama.

**Investigation:** Kay Sakuma, Risa Ishida, Yoshitake Takada.

**Methodology:** Kay Sakuma.

**Project administration:** Yoshitake Takada.

**Supervision:** Kay Sakuma, Yoshitake Takada.

**Visualization:** Kay Sakuma.

**Writing – original draft:** Kay Sakuma.

**Writing – review & editing:** Taketoshi Kodama.

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
