## [Decision Letter · Decision Letter 0]

31 Jul 2019

PONE-D-19-16483

Reconstructing the population history of the sandy beach amphipod Haustorioides japonicus using the calibration of demographic transition (CDT) approach

PLOS ONE

Dear Dr. Sakuma,

Thank you for submitting your manuscript to PLOS ONE. After careful consideration, we feel that it has merit but does not fully meet PLOS ONE’s publication criteria as it currently stands. Therefore, we invite you to submit a revised version of the manuscript that addresses the points raised during the review process.

We would appreciate receiving your revised manuscript by Sep 14 2019 11:59PM. To enhance the reproducibility of your results, we recommend that if applicable you deposit your laboratory protocols in protocols.io, where a protocol can be assigned its own identifier (DOI) such that it can be cited independently in the future. For instructions see: http://journals.plos.org/plosone/s/submission-guidelines#loc-laboratory-protocols

We look forward to receiving your revised manuscript.

Kind regards,

Tzen-Yuh Chiang

Academic Editor

PLOS ONE

Journal Requirements:

 [This study was partly supported by “A project for development of assessment methods for the coastal environment” funded by the Ministry of Agriculture, Forestry and Fisheries, and Grants-in-Aid for Scientific Research from the Ministry of Education, Culture, Sports, Science and Technology, Japan (25340114 and 15H02265).]. 

3. We note that [Figure 1] in your submission contains a map image which may be copyrighted. All PLOS content is published under the Creative Commons Attribution License (CC BY 4.0), which means that the manuscript, images, and Supporting Information files will be freely available online, and any third party is permitted to access, download, copy, distribute, and use these materials in any way, even commercially, with proper attribution. For these reasons, we cannot publish previously copyrighted maps or satellite images created using proprietary data, such as Google software (Google Maps, Street View, and Earth). For more information, see our copyright guidelines: http://journals.plos.org/plosone/s/licenses-and-copyright.

You may seek permission from the original copyright holder of Figure(s) [1] to publish the content specifically under the CC BY 4.0 license. 

If you are unable to obtain permission from the original copyright holder to publish these figures under the CC BY 4.0 license or if the copyright holder’s requirements are incompatible with the CC BY 4.0 license, please either i) remove the figure or ii) supply a replacement figure that complies with the CC BY 4.0 license. Please check copyright information on all replacement figures and update the figure caption with source information. If applicable, please specify in the figure caption text when a figure is similar but not identical to the original image and is therefore for illustrative purposes only.

4. We noted in your submission details that a portion of your manuscript may have been presented or published elsewhere. [Some nucleotide sequences (see Table S1) used in the present study were published previously and available online. The aim of the present study and thus analytical procedures are different from that of the previous one and also, we used additional sequences. This submission therefore does not constitute dual publication.]

* Please clarify whether this  publication was peer-reviewed and formally published. If this work was previously peer-reviewed and published, in the cover letter please provide the reason that this work does not constitute dual publication and should be included in the current manuscript.

Reviewers' comments:

Reviewer's Responses to Questions

**Comments to the Author**

1. Is the manuscript technically sound, and do the data support the conclusions?

Reviewer #1: Yes

Reviewer #2: Yes

2. Has the statistical analysis been performed appropriately and rigorously? 

Reviewer #1: Yes

Reviewer #2: Yes

3. Have the authors made all data underlying the findings in their manuscript fully available?

Reviewer #1: Yes

Reviewer #2: Yes

4. Is the manuscript presented in an intelligible fashion and written in standard English?

Reviewer #1: Yes

Reviewer #2: Yes

5. Review Comments to the Author

Reviewer #1: Sakuma and colleagues have reconstructed the demographic history of several populations of the amphipod Haustorioides japonicus using an approach based on demographic transition from an ancient epoch of constant population size to a modern epoch of rapid increase. The rapid population increase is assumed to have been triggered by warming temperatures following the Last Glacial Maximum. This approach is very interesting and has rarely been explored. As such, the present study is important as it explores the strengths and weaknesses of the method. The manuscript is well written, concise, the language is very good and the figures are of good quality. Overall, the analyses are done properly, although some clarifications are needed here and there. I have some remarks that should improve the presentation of the paper.

Line 41 “change of rate temperature” should be “rate of temperature change”.

Line 147. I recommend the authors to also use the R2 test (implemented in DnaSP) since it is more sensitive, especially regarding small sample sizes (Ramos-Onsins, S.E., Rozas, J., 2002. Statistical properties of new neutrality tests against population growth. Mol. Biol. Evol. 19, 2092–2100).

Line 154. “we built TEMs based on the whole sequence of the population to which the Tohoku population was assigned”. This needs to be rephrased as it is not clear. Does it mean that TEMs were build using all individuals from the NWP populations?

Line 156. Why did the authors not consider a strict clock as well for the BF comparisons?

Line 163. “9000 mutation values from the best model were obtained”. Please specify exactly to what parameter this refers to from the BEAST output file.

Lines 165-169. Please give more details here. What packages and their versions were used for these analyses?

Line 171. “inferred the history” should be “inferred the demographic history”. Write what BSP means when mentioned the first time. It is a bit confusing because the authors mention that they inferred the history of the Tohoku population, when in fact they analyzed this population together with the other NWP populations. This is made clear only in the results, but it should be clear here as well. Also in the BSP analyses it is not clear what kind of clock was used (strict vs relaxed). The authors mention the strict clock only in the second model based on the 1% evolutionary rate.

The authors compare their obtained rate with a phylogenetic (interspecific) rate of 1%. Although I see the use of such comparison, I do not see the point of the arbitrarily chosen 1% rate. I would recommend the authors to compare their intraspecific rate with interspecific (phylogenetic) rates that have been inferred for other crustaceans (e.g. Knowlton & Weigt 1998 https://doi.org/10.1098/rspb.1998.0568) and recently even amphipods (Copilas-Ciocianu et al. 2019 https://doi.org/10.1007/s13127-019-00401-7).

The authors should also clarify what they mean by “rate”. There is substitution rate and divergence rate (which is 2x substitution rate). In their MS, I suppose that the authors refer to the substitution rate. There is confusion in the literature about this, and that is why there should be more clarity (Schenekar & Weiss 2011 doi:10.1038/hdy.2011.48).

Line 325. Here the authors should also mention that several studies have shown fast intraspecific rates in crustaceans as well (Audzijonyte&Vainola 2006 http://dx.doi.org/10.1111/j.1365-294X.2006; Crandall et al. 2012 http://dx.doi.org/10.1093/molbev/msr227).

Reviewer #2: I found your work on the sandy beach amphipod Haustorioides japonicus really interesting. It represents an important and interesting contribution as it shows, for the first time, the power of the calibration of demographic transition approach and the importance of using an adequate evolutionary rate in an amphipod species.

In general, the organization of the manuscript is satisfactory and its easy reading. The TITLE clearly reflects the contents. The ABSTRACT is sufficiently informative and has a correct length. The INTRODUCTION is very interesting, clear and concise, and contains enough background to put the reader in context. The statement of the objective is adequate and appropriate. MATERIAL AND METHODS are clearly explained, being sufficiently informative to allow replication. The analyses are properly detailed, and they are solid. The RESULTS are clearly presented and described in a logical order, and their interpretation and posterior discussion are justified by the data and consistent with the objectives. The DISCUSSION and CONCLUSIONS are very clear and well organized. Authors explain properly their results obtained with many adequate references. Finally, the English is adequate.

Overall, I think the paper is interesting and worthy of publication. I think that this paper could be accepted in PLOS ONE as the subject of the manuscript falls within the scope of the journal. Some suggestions and corrections were made to improve it (see below) that should be relatively easy for the authors to fix. So, I consider that this manuscript is acceptable for publication after minor revisions.

IMPORTANT:

According to the “World Register of Marine Species” (WoRMS) database, Haustorioides japonicus is an unaccepted species. The accepted species name corresponds to “Eohaustorioides japonicus (Kamihira, 1977)”. Therefore, the authors must correct this along the whole manuscript including the corresponding title and the abstract.

INTRODUCTION

- Line 86: “(Dogielinotidae: Amphioda)”. The Order name must be before the Family name. Therefore, the authors should place “Amphipoda” before “Dogielinotidae”.

- Line 95: “genetically distinct”. Genetically distinct from what? The authors should clarify this to avoid confusion.

- Line 97: “BSP”. What does BSP mean? Bayesian Skyline Plot? When you use an abbreviation, its meaning should be specified the first time it appears in the manuscript. Therefore, the authors should include this information.

- Line 97: “genealogy”. Please, include “of the sandy amphipod Eohaustorioides japonicus”, just to make it clear.

MATERIALS AND METHODS

- Line 109 - 110: “three sandy beaches in the Tohoku region and two sites along the coast of Hokkaido”. In the present study, according to Figure 1 legend (empty circles) and locations in Table 1, only one location was sampled along the coast of Hokkaido (site 4, Usu). Therefore, the authors should replace “two sites along…” with “one site along…”

- Line 111: “parts”. Which parts were preserved? Did the authors preserve some specific parts for some reason? Please, specify this.

- Line 115: Please, include a “a” after “using”.

- Line 135: I will include a sentence saying that sequences were deposited in GenBank (accession numbers: LC474498–LC474506) (S1 Table).

- Line 137: “The 300 sequences”. Sequences used by Takada et al. [13] were 105 and not 300, as the authors can see in the Table S1 of Takada et al.’s paper. The authors should correct this.

RESULTS

- Line 207: “(site 7)”. The authors should include “Fig 1” after “(site 7)” to ensure that readers understand exactly which location is talking about.

- Line 211: “two populations”. Really, NWP is not only one population but a group of them. Therefore, I think, the authors should replace “two populations” with “two group of populations”. The same should be modified in Table 2 captions.

DISCUSSION

- Line 289: “Littorina saxatilis”. The authors should include the authority of this taxon.

REFERENCES:

- Line 412: Please, remove the comma after “Kamihira”

- Line 418: Please, remove the comma after “Kamihira”

- Line 465: “Haustorioides japonicus” must be in italics. Please, change it.

FIGURES AND TABLES:

- Figure 1: It is clear and appropriate for the data being presented. However, I think, it is necessary to include/specify two things:

o According to the manuscript, Tohoku region is located along the Pacific coast of northeastern Honshu Island, Japan. However, this region is not clearly indicated in the Figure. As this region is very important in the present study, it should be clearly indicated to put the readers in context and to ensure that they understand exactly which region corresponds to Tohoku region.

o This Figure is extremely similar to Figure 1 of Takamada et al. [13]. In fact, the maps used are the same. I think, the authors should include in the legend something like: “Modified from Takada et al. [13]”.

- Figure 2: In the Figure 2 (A) appears a total of 16 haplotypes. However, in Table S1 and in the manuscript (for example, line 194) is stated the existence of 15 haplotypes, not 16. Specifying, the haplotype Hja_033 appears in Figure 2 but is missing in Table S1. Could the authors explain this fact? Please, correct this properly in the paper.

- Table 1: Please, include “(n)” after “sample size”

- Table 2: Please, replace “two populations” with “two group of populations” in Table legend. See comments for Line 211.

- S1 Table: “total number (n)”. Total number of individuals? Please, indicate this in the Table caption.

6. PLOS authors have the option to publish the peer review history of their article (what does this mean?). If published, this will include your full peer review and any attached files.

Reviewer #1: Yes: Denis Copilas-Ciocianu

Reviewer #2: Yes: M. Pilar Cabezas

---

## [Author Response · Author response to Decision Letter 0]

11 Sep 2019

Response to Reviewer #1

Phylogenetic substitution rate used in the comparison. The reviewer suggested to use inter- or intraspecific rate for other crustaceans such as 0.7% (Knowlton &Weight 1998) and 1.77% (Copilas-Ciocianu et al. 2019), instead of “general” evolutionary rate of 1% we used in the comparison. It is true that the rate of 1% is not estimated for crustacean species and we have reanalyzed the data with 0.7%/my by Knowlton &Weight (1998). The point in the figure 4 (and 5) is the disparity between general (or easy) rate and CDT rate, and also an uncertainty which is properly expressed by using stochastic clock model. Because showing the results with three (or more) different rates would increase the complexity of the figure (please check attached file Fig3-2.pdf), we would like to show BSPs with two rates. Ultimately a choice of the clock is arbitrary process and it is out of the focus.

Line 41: The statement has changed following the reviewer’s comment.

Line 153: We performed R2 test according to the reviewer’s suggestion and have added the statement regarding the analysis in Materials and methods (Line 153) and Results (Line 220 and Table 2) sections.

Line 159: We have added the statement that “(Hereafter ‘rate’ means substitution rate, not ‘divergence’ rate)” following the reviewer’s suggestion that it should be clarified what the rate means (substitution rate or divergence rate).

Line 160: As the reviewer suggested, “TEMs were built using all individuals from the NWP populations” is just what we mean. We however do not know to which of the five populations the Tohoku population assigned at this point. The statement has therefore been rephrased as follows: “we built TEMs using all individuals from Tohoku as well as its parental population among currently known sandy beach amphipod populations.”.

Lines 163, 231, 243: We have included the result for the strict clocks following the reviewer’s comment, while the result for the exponential TEM with strict clock model is not shown because the run did not converge.

Line 169: “9000 mutation values” has been replaced with “9000 values of the mutation-scaled transition time” for clarity.

Lines 175: We used ‘fitdistr’ function in the MASS v7.3 package, and we have included the name of the package in Line 175.

Line 99, 179, 252: Following the reviewer’s comment, ”BSP” in Line 99 has been spelled out as “Bayesian Skyline Plot” and reference added. Then the statement in Line 179 has been rephrased as follows: “We inferred the demographic history of the Tohoku and its parental population using a BSP.” For the clock model used in the BSP analysis, we stated that “The molecular rate was modeled based on the statistical distribution from CDT.” in Line 183, because the model is not determined at this point. For better clarity, we have replaced the phrase with “The molecular rate was modeled following the best clock model and statistical distribution from CDT”, also added the statement in Result section (Line 255) to show that exponential relaxed clock was applied to BSP.

Line 340: We have added the references for the faster evolutionary rate of crustacean species according to the reviewer’s suggestion. The point is the rate variation among crustacean taxa and we do not discuss each rate here and just introduced them as examples.

Response to Reviewer #2

Scientific name of the sandy beach amphipod. As suggested by the reviewer, generic name of ‘Eohaustorioides’ is also known for a single species of Haustorioides japonicus Kamihira 1977, although it seems not to be widely accepted. The genus Euhaustrioides was originally introduced in a book chapter without detailed description (Bousfield and Tzvetkova 1982), and the record of WoRMS is solely based on this information. Later Jo (1988) and Kamihira (1999), the authority of this amphipod group, refuted the establishment of Eohaustorioides, and the status of the genus seems still controversial. Our research team is now preparing a taxonomic paper with additional refutation on using this name. Furthermore, WoRMS is a web database and has nothing to do with scientific justification, while it argues that the database is controlled by “taxonomic experts”. We therefore avoided using Eohaustorioides in this manuscript in the first round. If it is further required to use Eohaustorioides in this manuscript, we would like to follow the editor/reviewer’s suggestion.

Line 86: We have corrected the statement following the reviewer’s suggestion.

Line 95: We have rephrased the statement as “genetically distinct from other known populations” following the reviewer’s comment.

Line 97, 98: “BSP” has been spelled out as “Bayesian Skyline Plot”, and the phrase “of the sandy amphipod Haustorioides japonicus” has been added after “genealogy”, according to the reviewer’s suggestion.

Line 111: “two sites” has been replaced with “one site”.

Line 116: “a” is inserted after “using” as suggested.

Line 142: GenBank accession numbers are added following the reviewer’s recommendation.

Line 143: Number of sequences are corrected.

Line 216: We have added “Fig 1” after “site 7” following the suggestion. 

Lines 219, 223: The phrase “two populations” was replaced with “two group of populations”.

Line 303: The authority is added after the name of the species.

Lines 428, 434, 488: Formatting mistakes in the Reference section has been corrected according to the reviewer’s comments.

Figure 1 has been revised following the reviewer’s comment.

S1 Table and Figure 2: Haplotype Hja_033 has been removed from the figure because it is haplotype from the different genetic population (SJN, northern Sea of Japan) and not included in the analysis. The caption in the Table has been revised to show that “(n)” means total number of individuals.

Table 1 (Line 129): “(n)” is included after “sample size”.

---

## [Decision Letter · Decision Letter 1]

25 Sep 2019

Reconstructing the population history of the sandy beach amphipod Haustorioides japonicus using the calibration of demographic transition (CDT) approach

PONE-D-19-16483R1

Dear Dr. Sakuma,

We are pleased to inform you that your manuscript has been judged scientifically suitable for publication and will be formally accepted for publication once it complies with all outstanding technical requirements.

With kind regards,

Tzen-Yuh Chiang

Academic Editor

PLOS ONE

Additional Editor Comments (optional):

Reviewers' comments:

Reviewer's Responses to Questions

**Comments to the Author**

1. If the authors have adequately addressed your comments raised in a previous round of review and you feel that this manuscript is now acceptable for publication, you may indicate that here to bypass the “Comments to the Author” section, enter your conflict of interest statement in the “Confidential to Editor” section, and submit your "Accept" recommendation.

Reviewer #1: All comments have been addressed

Reviewer #2: All comments have been addressed

2. Is the manuscript technically sound, and do the data support the conclusions?

Reviewer #1: Yes

Reviewer #2: Yes

3. Has the statistical analysis been performed appropriately and rigorously? 

Reviewer #1: Yes

Reviewer #2: Yes

4. Have the authors made all data underlying the findings in their manuscript fully available?

Reviewer #1: Yes

Reviewer #2: Yes

5. Is the manuscript presented in an intelligible fashion and written in standard English?

Reviewer #1: Yes

Reviewer #2: Yes

6. Review Comments to the Author

Reviewer #1: (No Response)

Reviewer #2: Authors perfectly addressed the comments made by the two reviewers. The manuscript is now acceptable for publication.

7. PLOS authors have the option to publish the peer review history of their article (what does this mean?). If published, this will include your full peer review and any attached files.

Reviewer #1: Yes: Denis Copilas-Ciocianu

Reviewer #2: Yes: M. Pilar Cabezas

---

## [Editor Report · Acceptance letter]

30 Sep 2019

PONE-D-19-16483R1 

Reconstructing the population history of the sandy beach amphipod *Haustorioides japonicus* using the calibration of demographic transition (CDT) approach 

Dear Dr. Sakuma:

I am pleased to inform you that your manuscript has been deemed suitable for publication in PLOS ONE. Congratulations! Your manuscript is now with our production department. 

With kind regards,

on behalf of

Dr. Tzen-Yuh Chiang 

Academic Editor

PLOS ONE